# A Novel Model Fusion Approach for Greenhouse Crop Yield Prediction

**Liyun Gong** [1], **Miao Yu** [1,*], **Vassilis Cutsuridis** [1], **Stefanos Kollias** [1] and **Simon Pearson** [2]

1   School of Computer Science, University of Lincoln, Lincoln LN6 7TS, UK
2   Lincoln Institute for Agri-Food Technology, University of Lincoln, Lincoln LN6 7TS, UK
*   Correspondence: myu@lincoln.ac.uk

**Abstract:** In this work, we have proposed a novel methodology for greenhouse tomato yield prediction, which is based on a hybrid of an explanatory biophysical model—the Tomgro model, and a machine learning model called CNN-RNN. The Tomgro and CNN-RNN models are calibrated/trained for predicting tomato yields while different fusion approaches (linear, Bayesian, neural network, random forest and gradient boosting) are exploited for fusing the prediction result of individual models for obtaining the final prediction results. The experimental results have shown that the model fusion approach achieves more accurate prediction results than the explanatory biophysical model or the machine learning model. Moreover, out of different model fusion approaches, the neural network one produced the most accurate tomato prediction results, with means and standard deviations of root mean square error (RMSE), r2-coefficient, Nash-Sutcliffe efficiency (NSE) and percent bias (PBIAS) being $17.69 \pm 3.47$ g/m$^2$, $0.9995 \pm 0.0002$, $0.9989 \pm 0.0004$ and $0.1791 \pm 0.6837$, respectively.

**Keywords:** biophysical model; deep neural network; recurrent neural network; convolutional neural network; model fusion; crop yield prediction

## 1. Introduction

Crop yield prediction is important for managing greenhouse crop growth. The greenhouse crop yield prediction results can be exploited by cultivators and farmers to make more appropriate greenhouse management plans and financial decisions. Moreover, crop yield prediction is also an important component integrated into a greenhouse control system, which facilitates finding the optimal control parameter settings to guarantee the maximum greenhouse crop yield [1]. However, crop yield prediction is an extremely challenging task. As shown in [2,3], crop yield prediction is dependent on a variety of factors (e.g., temperature, carbon dioxide concentrations, radiation, etc.), and it is not straightforward to construct an explicit model to reflect the relationship between these factors and crop yield.

Although there are many research works related to crop yield prediction for open farming field scenarios, a relatively small amount of work focuses on greenhouse crop yield prediction, which can be divided into two main categories: the explanatory biophysical model-based approach and the data-driven/machine learning model-based approach.

The biophysical model-based approach predicts the crop yield using a series of ordinary differential equations (ODEs) describing the relationship between environmental factors and crop growth. For example, the reduced Tomgro model proposed by Jones et al. modeled the tomato growth and fruit yield based on environmental information, such as temperature, solar radiation and $CO_2$ concentration, inside a greenhouse [4]. In [5], different optimization algorithms have been compared for the calibration of the reduced Tomgro model. The results have shown that the particle swarm optimization (PSO) algorithm can achieve the best model calibration performance. Compared with the simplified, reduced Tomgro model, a more complex Tomsim biophysical model was proposed in [6], which

contained multiple sub-modules modeling different aspects related to tomato growth, including photosynthesis, dry matter production, truss appearance rate, fruit growth period and dry matter partitioning. A biophysical model that describes the effects of greenhouse climate parameters on crop yield was proposed in [2]. The results demonstrated that the tomato yield could be successfully simulated at different geographical locations under different temperatures, light and $CO_2$ conditions. In [7], an integrated yield prediction model was proposed to integrate both the Tomgro model [3] and the Vanthoor model [2] for predicting the greenhouse crop yield based on controllable greenhouse climate parameters. A variety of biophysical models (e.g., the Vanthoor model [2], the Tomsim model [6] and the Greenhouse Technology applications (GTA) model [8]) and their combined versions were compared in [9]. Experimental studies showed that the combined version can outperform the original models for yield prediction with smaller root mean square errors (RMSEs).

The biophysical model is bio-physically meaningful and explainable by reflecting the actual growth process of crops. However, the biophysical model is described by ODEs, which cannot fully reflect the complex biophysical process during crop growth. Moreover, there are many associated intrinsic model parameters, and the crop prediction accuracy is highly sensitive to such model parameters [10]. The parameters setting suitable for predicting greenhouse crop yields in one region may not be workable for other regions [10]. These limitations adversely affect the biophysical model performance for yield prediction.

The machine learning model-based approach models the crop yield output as a complex/nonlinear function of the greenhouse climate and historical growth information. For example, linear and polynomial regression models were used in [11] for strawberry growth and fruit yield based on environmental factors (e.g., average daily air temperature (ADAT), relative humidity (RH) and soil moisture content (SMC)). However, a linear or polynomial relationship is not always valid. In [12], ANN was exploited for the pepper fruit yield prediction based on fruit water content, days to flowering initiation and other relevant factors. Variants of classical ANN have also been applied for yield prediction. An Evolving Fuzzy Neural Network (EFuNN) was proposed in [13] for tomato yield prediction based on different greenhouse environmental variables (e.g., temperature, $CO_2$, vapor pressure deficit (VPD), historical yield, etc.). A dynamic artificial neural network (DANN) [14] was developed for tomato yield prediction based on different factors (e.g., $CO_2$, transpiration, radiation, historical yield, etc.). Recently, deep learning technology has also begun to be applied for greenhouse crop yield prediction. In [15,16], researchers adopted a recurrent neural network (RNN) model for predicting tomato and ficus yields. The evaluation results showed that deep learning-based approaches outperformed traditional machine learning algorithms with lower root mean square errors (RMSEs). A hybrid of the temporal convolutional network (TCN) and recurrent neural network (RNN) models for the crop yield prediction was employed in [17], which showed that this hybrid model-based approach can achieve higher prediction accuracy over both the classical machine learning-based approaches and exploiting the TCN or LSTM model solely.

As the machine learning-based approaches are data-driven ones, the performance of the machine learning-based approaches for yield prediction heavily depends on the amount/quality of the data used for training the machine learning model. The trained model would be less accurate if the training data are scarce or of poor quality. Moreover, the machine learning-based approaches can suffer from the 'overfitting' problem [18], leading to poor yield prediction performance for new scenarios.

Considering the limitations existing in both explanatory and machine learning models, it is challenging to always predict crop growth accurately by relying solely on an explanatory model or a machine learning one. In this work, we have developed model fusion approaches to combine crop yield outputs from two different types of models to generate the final yield prediction outcome. In this way, the limitations of each model will be compensated for achieving a more accurate yield prediction result. Specifically, we have exploited both a biophysical model, called the reduced Tomgro model, and a CNN-RNN machine learning model for tomato yield prediction. The tomato yield prediction results

based on the two models are further fused by another 'fusion model' to generate the final yield prediction outcome. From the experimental studies, it has been shown that the performance of the model fusion approach achieves higher performance than solely using the biophysical model or machine learning model for yield prediction.

## 2. Methodology

In this work, we propose a fusion-based model of crop yield prediction for predicting the tomato yield in a greenhouse, which is illustrated in Figure 1. The proposed approach is a fusion of a biophysical model called a reduced Tomgro model in [4] and a CNN-RNN machine learning model. The prediction results from both models are fused together in a fusion module to generate the final prediction output. Different parts of the diagram will be elaborated on in the next sub-sections.

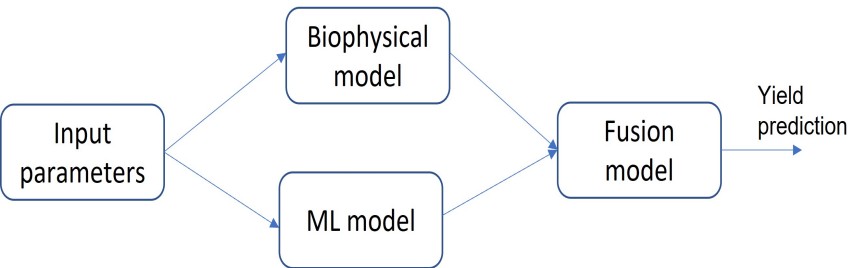

**Figure 1.** The sketch diagram of the model fusion approach for yield prediction.

### 2.1. Biophysical Model

The biophysical model used in this work is a reduced state-variable tomato model [4], which is used for modeling the dynamics of the total dry matter production and distribution in fruit and mature, dry weight based on greenhouse climate parameters, including the photosynthetically active radiation (PAR) in (mmol/m$^2$/s), air temperature (°C) and $CO_2$ concentration (ppm), depending on photosynthesis and respiration processes. This model includes the number of mainstem nodes ($N$), leaf area index ($LAI$), total plant weight($W$), fruit weight($W_F$) and mature fruit dry weight ($W_M$) as the state variables, whose evolutions with respect to time are modeled by ODEs. Compared with the more complex model as the TomSim model in [6], the reduced Tomgro model simulates the same photosynthesis, respiration and development process but with new leaf area and dry matter growth relationships being developed.

In specific, the main ODEs for modeling the evolutions of state variables in the reduced state-variable Tomgro model are shown below as in [4]:

$$\frac{dN}{dt} = N_m \cdot f_N(T) \tag{1}$$

$$\frac{d(LAI)}{dt} = \rho t \delta \cdot \lambda(T_d) \frac{exp(\beta \cdot (N - N_b))}{1 + exp(\beta \cdot (N - b))} \cdot \frac{dN}{dt} \tag{2}$$

$$\frac{dW}{dt} = \frac{dW_F}{dt} + (V_{max} - p_1) \cdot \rho \cdot \frac{dN}{dt} \tag{3}$$

$$\frac{dW_F}{dt} = GR_{net} \cdot \alpha_F \cdot f(T_d) \cdot (1 - exp(-v(N - NFF))) \cdot g(T_{daytime}) \tag{4}$$

$$\frac{dW_M}{dt} = D_F(T_d) \cdot (W_F - W_M) \tag{5}$$

where $\frac{dN}{dt}$ represents the node development rate modeled as the multiplication of ($N_m$) representing the maximum daily rate of node appearance rate and a function $f_N(T)$ depending on the non-optimal temperature. The update rate of the $LAI$ in (2) is dependent on both the node number $N$ and the daily temperature $T_d$. $W$ in (3) represents the total above-ground dry weight, which is calculated as a weighted sum of the fruit growth rate

$\frac{dW_F}{dt}$ and node growth rate $\frac{dN}{dt}$. The fruit growth starts when the node number reaches $NFF$ and increases asymptotically to a maximum value, as indicated in (4), while the fruit growth rate is dependent on the temperature $T_d$, $T_{daytime}$ and $GR_{net}$ functions, as in [4]. $\frac{dW_F}{dt}$ in (5) represents the mature fruit dry matter development rate, which indicates the rate that fruit grow to mature stages. It depends on the daily average temperature $T_d$, fruit weight $W_F$ and mature fruit weight $W_M$. More details on parameter descriptions in the reduced Tomgro model can be found in [4].

The parameters of the reduced Tomrgo model can be calibrated by different algorithms (e.g., genetic algorithm (GA), differential evolutionary (DE) or particle swarm optimization, as in [5]). The calibrated Tomgro model can then be applied for the tomato yield prediction by using the related OEDs to calculate $W$, $W_F$ and $W_M$ values.

Compared with models in [2,6,7], the reduced Tomgro model is light-weight and has much fewer model parameters to tune, making it more convenient to be calibrated for real-site tomato growth modeling, which has been successfully applied to real-site tomato yield predictions as reported in multiple research works [10,14,19].

### 2.2. ML Model

A CNN-RNN-based machine learning approach is applied for tomato yield prediction, which is a combination of a convolutional neural network (CNN) and a recurrent neural network (RNN). The structure of the CNN-RNN is shown in Figure 2. The CNN part (composed of multiple blocks of 1D convolutional layers (indicated as $^*N$ in Figure 2), weight norm layers and ReLU/Drop operations) extracts the most representative spatial features from the original normalized input temporal sequence, while the extracted features are fed into an RNN containing multiple LSTM units to effectively capture the temporal dependencies for the final yield prediction. By exploiting the CNN-RNN, both spatial and temporal information from the original input sequence is fully exploited to achieve accurate yield prediction results.

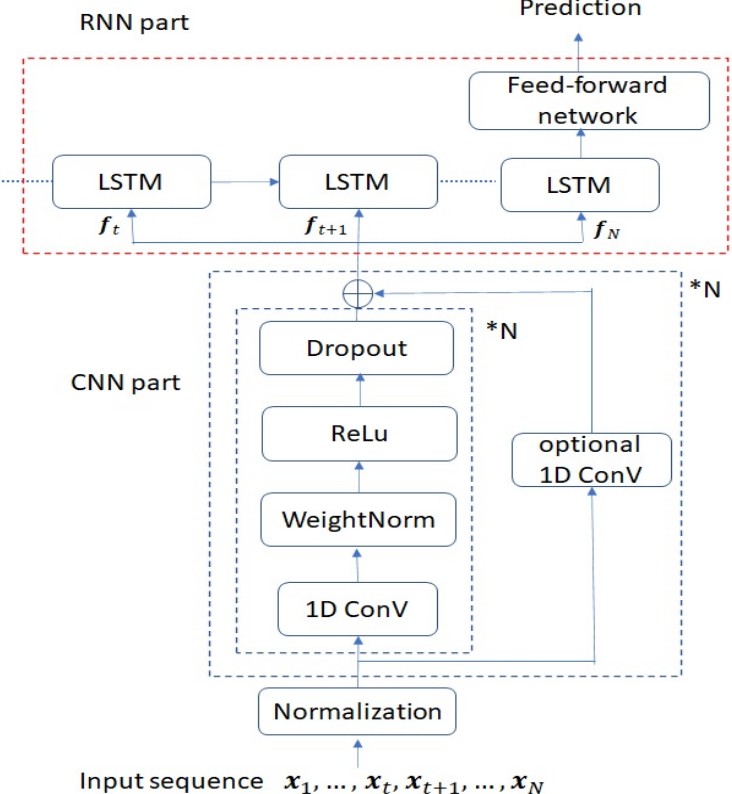

**Figure 2.** The diagram of the CNN-RNN model for crop yield prediction.

### 2.2.1. Data Normalization

A temporal sequence denoted as $x_1, \ldots, x_N$ is taken as the network input, where $x_t$ represents a vector containing different factors recorded at the time instance $t$, including historical yield information (g/m$^2$), greenhouse $CO_2$ concentration (ppm), greenhouse temperature (°C), humidity deficit (g/kg), relative humidity (percentage) and radiation (W/m$^2$).

Normalization is first applied to the data to normalize each factor to a range of [0, 1] by the following equation:

$$\hat{x}_t^i = \frac{x_t^i - x_{min}^i}{x_{max}^i - x_{min}^i} \tag{6}$$

where $x_t^i$ represents the $i$-th factor at the time step $t$ while $x_{min}^i$ and $x_{max}^i$ represent the corresponding maximum and minimum values for the related factor.

### 2.2.2. CNN Part

The normalized input temporal sequence is then fed into a CNN part for representative feature extraction. As in Figure 2, the CNN part contains multiple blocks, while each block contains a series of 1D convolutional operations, weight norm operations, ReLU/Drop operations and an optional 1D convolutional operation. The 1D convolution is the main operation in the CNN part. As in [20], the 1D convolution performed on a temporal sequence, $I_t$ is calculated as:

$$O_t^k = f(\sum_{i,j} w_k^{i,j} I_{t+i}^j + b) \tag{7}$$

where $O_t^k$ represents the k-th output at time instance $t$. $I_t^j$ represents the j-th element of $I_t$. $w_k^{i,j}$ and $b$ represent the convolutional kernel weight and bias, respectively. $f(\cdot)$ represents an activation function where ReLU is used in this work. As in [21], the weight normalization re-parameterizes the 1D convolution weights into the vector and scalar parameters for accelerating the network training. A certain percentage of weights during the network training will be dropped to improve the generalization performance.

High-level features are extracted from the original normalized input sequence based on a series of 1D convolutions in the CNN part. Moreover, as in Figure 2, a residual link containing one optional 1D convolution operation is applied to extract low-level features, which are added up to high-level features to obtain the final feature set (denoted as $f_1, \ldots, f_N$ in the figure), integrating the comprehensive high and low-level feature information.

### 2.3. RNN Part

The extracted feature sequence from the CNN is then fed into an RNN model containing multiple LSM units. The RNN is exploited to model/capture the temporal dependencies between the extracted feature sequence used for yield prediction. However, the traditional RNN problems exist of both gradient vanishing and gradient explosion [22]. Currently, the most popular way to overcome the limitations of the traditional RNN is to adopt a new architecture incorporating long short-term memory (LSTM) units, known as LSTM-RNN [23], which is used in our work. In each LSTM unit, there is a series of arithmetic operations defined in [23] as below:

$$i_t = \sigma(W_{xi}x_t + W_{hi}h_{t-1} + W_{ci}c_{t-1} + b_i)$$
$$f_t = \sigma(W_{xf}x_t + W_{hf}h_{t-1} + W_{cf}c_{t-1} + b_f)$$
$$c_t = f_tc_{t-1} + i_t tanh(W_{xc}x_t + W_{hc}h_{t-1} + b_c)$$
$$o_t = \sigma(W_{xo}x_t + W_{ho}h_{t-1} + W_{co}c_{t-1} + b_o)$$
$$h_t = o_t tanh(c_t)$$

(8)

where $x_t$, $h_t$ and $o_t$ represent the LSTM input, LSTM state and LSTM output at time instance $t$. $c_t$ is the LSTM cell value representing encoded historical information obtained from previous data samples before $t$. $\sigma(\cdot)$ and $tanh(\cdot)$ represent sigmoid and tanh functions. The other parameters represent weights and biases associated with an LSTM unit.

The LSTM-RNN model encodes the whole input sequence into a single feature vector (obtained from the last LSTM unit's outputs, as shown in Figure 2), which contains the information of the whole sequence incorporating the temporal dependency information between consecutive time instances. Finally, the extracted feature vector is then fed into a simple forward neural network model for yield prediction.

*2.4. Fusion Model*

The outputs from the biophysical model and machine learning model are fed into a fusion module for generating the final crop yield prediction output, as represented in the following equation

$$out_{fusion} = F(out_{bio}, out_{ml})$$

(9)

where $F(\cdot)$ represents a fusion model. $out_{bio}$ and $out_{ml}$ represent the outputs from the reduced Tomgro model and machine learning model, respectively, while $out_{fusion}$ represents the final yield prediction output generated from the fusion model. In this work, we have explored different $F(\cdot)$ models for fusing, including the linear model, Bayesian model, neural network model, random forest model and gradient-boosting model. Comparisons of the different fusion models are shown in the experimental section.

**3. Experimental Studies**

The proposed model fusion-based greenhouse tomato yield prediction approach is evaluated on two years of tomato growth datasets collected from a real tomato grower in the UK, including both the tomato yield and greenhouse environmental parameters (e.g., $CO_2$ concentration (ppm), greenhouse temperature (°C), humidity deficit (g/kg), relative humidity (percentage) and radiation (W/m$^w$)) from a period of two years. The data corresponding to the first year are used for training the biophysical model, machine learning model and the combined model, while the data for the second year are used for testing. Figure 3 visualizes different greenhouse environmental parameters for both datasets, while their statistics are shown in Table 1.

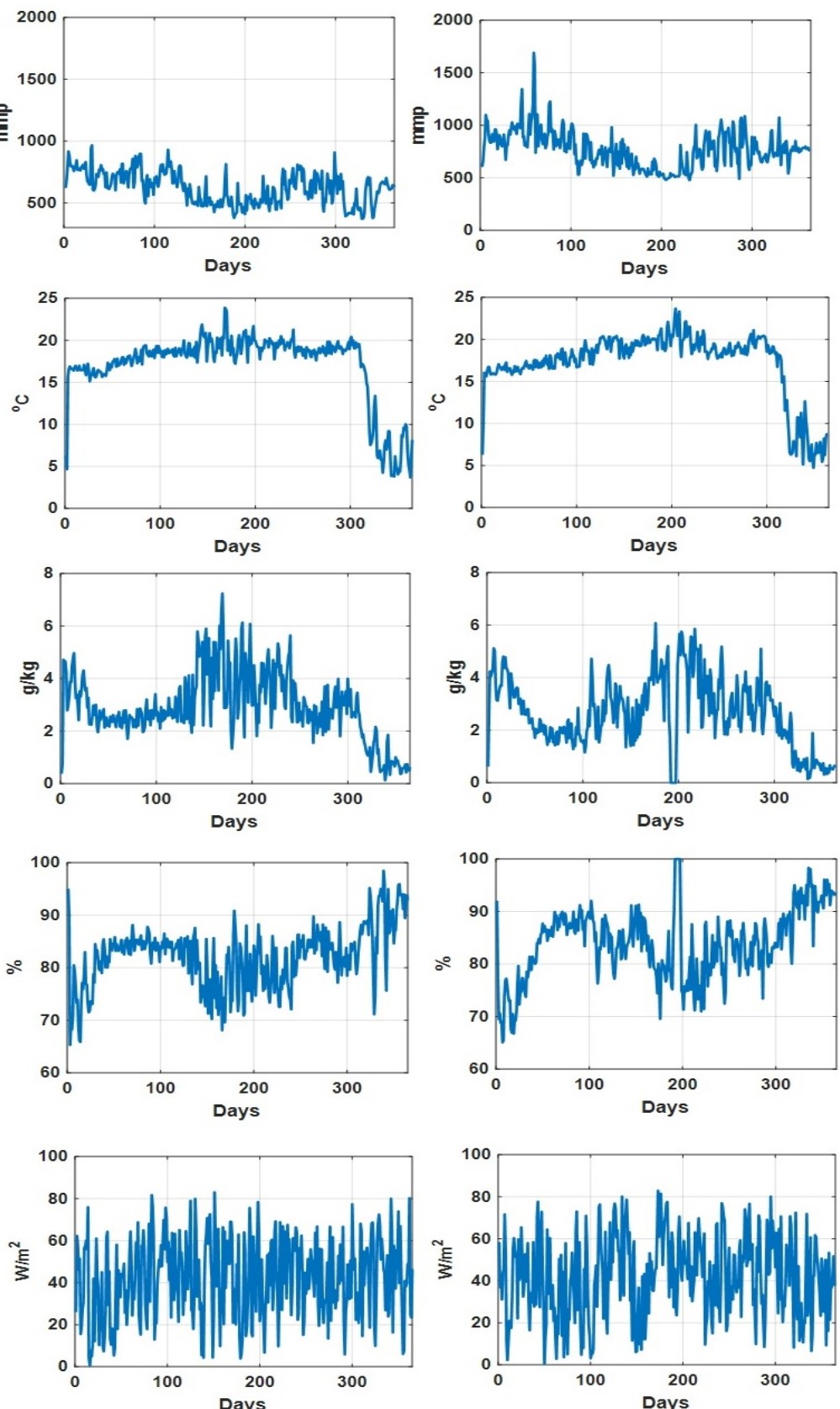

**Figure 3.** $CO_2$ concentration (mmp), temperature (°C), humidity deficit (g/kg), relative humidity (percentage) and radiation(W/m$^2$) for the training dataset (**left column**) and testing dataset (**right column**).

**Table 1.** Statistics of greenhouse environmental parameters for different datasets.

| | | Training Dataset | Testing Dataset |
|---|---|---|---|
| $CO_2$ (mmp) | Min | 370.94 | 478.05 |
| | Max | 967.40 | 1691.43 |
| | Median | 629.97 | 769.79 |
| | Mean | 624.19 | 770.37 |
| | Standard deviation | 129.58 | 175.61 |
| Temperature (°C) | Min | 3.68 | 4.72 |
| | Max | 23.89 | 23.69 |
| | Median | 18.46 | 18.31 |
| | Mean | 17.01 | 17.18 |
| | Standard deviation | 4.25 | 3.94 |
| Humidity deficit (g/kg) | Min | 0.13 | 0 |
| | Max | 7.27 | 6.08 |
| | Median | 2.78 | 2.58 |
| | Mean | 2.91 | 2.65 |
| | Standard deviation | 1.29 | 1.33 |
| Relative humidity (%) | Min | 65.31 | 65.09 |
| | Max | 98.50 | 100 |
| | Median | 83.22 | 84.73 |
| | Mean | 82.19 | 83.99 |
| | Standard deviation | 5.88 | 6.72 |
| Radiation (W/m$^2$) | Min | 0.58 | 0.59 |
| | Max | 83.02 | 82.91 |
| | Median | 43.41 | 42.81 |
| | Mean | 42.19 | 42.17 |
| | Standard deviation | 18.92 | 19.37 |

*3.1. Biophysical Model Results*

The reduced Tomgro model, as mentioned in Section 2.1, is calibrated to be applied for the tomato yield prediction. Specifically, Table 2 shows some of the key parameters of the reduced Tomgro model that need to be calibrated. We have exploited/compared three evolutionary algorithms, including GA, DE and PSO, for model calibration. The root mean square error (RMSE) between the recorded yields in the training dataset and predicted ones by the model is taken as the fitness function for all three algorithms.

**Table 2.** Key parameters for the reduced Tomgro model.

| Parameter | Description | Range of Estimate | Unit |
|:---:|:---:|:---:|:---:|
| $N_m$ | Maximum node development rate | [0.35, 0.4] | node d$^{-1}$ |
| $N_b$ | Parameter in expolinear equation | [14, 16] | node |
| $\delta$ | Maximum leaf area expansion | [0.05, 0.08] | m$^2$ node$^{-1}$ |
| $\beta$ | Parameter in expolinear equation | [0.45, 0.55] | node$^{-1}$ |
| $V_{max}$ | Maximum increase in vegetative tissue d.w. growth per node | [8, 10] | g[d.w.] node$^{-1}$ |
| $\tau$ | $CO_2$ coefficiency | [0.08, 0.12] | μmol m$^2$ s$^{-1}$ |
| $T_{crit}$ | Critic temperature | [19, 21] | °C |
| $v$ | Transition from vegetative development to fruit development | [0.8, 1] | node$^{-1}$ |
| $K$ | Development time from first fruit to ripe one | [0.8, 1] | node |
| $m$ | Light transmission coefficient | [0.01, 0.015] | dimensionless |
| $N_{FF}$ | Nodes per plant | [16, 18] | node |
| $\alpha_F$ | Maximum new growth to fruit partitioning | [0.8, 1] | [fraction] d$^{-1}$ |
| $E$ | Growth efficiency | [0.9, 1.2] | g[d.w.] g$^{-1}$ [CH$_2$O] |
| $D$ | $CO_2$ to $CH_2O$ conversion coefficient | [4, 6] | gm$^{-2}$h$^{-1}$ |

The evolutions of the RMSE fitness function values with respect to evolutionary algorithm iteration numbers are shown in Figure 4a. We can see that the obtained minimum fitness values all decrease and converge after a few iterations, indicating that good solutions to the model parameters are successfully obtained. The calibrated reduced Tomgro model is then applied to predict the tomato yield. As shown in Figure 4b, we can see that intuitively the predicted tomato yield is close to the ground truth ones.

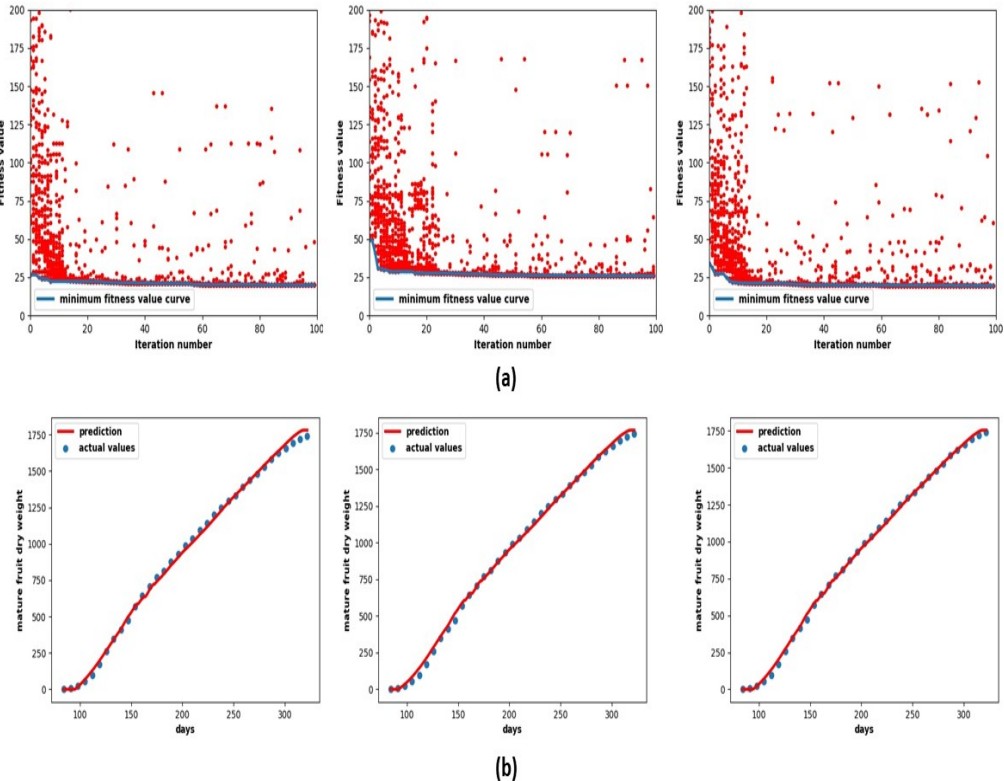

**Figure 4.** Reduced Tomgro model calibration by three optimization algorithms. (**a**) The fitness values of solution candidates (red dot) and minimum fitness value (blue line) for GA (**left**), DE (**middle**) and PSO (**right**). (**b**) The illustration of yield prediction by the reduced Tomgro model calibrated by GA (**left**), DE (**middle**) and PSO (**right**).

Quantitative analysis is performed to compare the performance of different evolutionary algorithms for calibrating the reduced Tomgro models for yield prediction. Multiple trials of evaluations are made and the obtained mean and standard deviation (std) of different metrics ( RMSEs, r2-coefficient, Nash-Sutcliffe efficiency (NSE) and percent bias (PBIAS)) calculated based on the testing dataset are summarized in Table 3. We can observe that the reduced Tomgro model calibrated by the PSO algorithm can achieve the minimum mean RMSE and absolute PBIAS and the maximum mean R2 and NSE, which indicates the most accurate yield prediction.

**Table 3.** Statistics of different metrics obtained by the reduced Tomgro model calibrated by different algorithms.

|  | GA | DE | PSO |
|---|---|---|---|
| Mean and std of RMSE (g/m$^2$) | 45.16 ± 13.11 | 55.36 ± 18.07 | 36.17 ± 7.64 |
| Mean and std of R2 | 0.9969 ± 0.0017 | 0.9953 ± 0.0030 | 0.9980 ± 0.0001 |
| Mean and std of NSE (g/m$^2$) | 0.9938 ± 0.0033 | 0.9907 ± 0.0060 | 0.9961 ± 0.0016 |
| Mean and std of PBIAS (g/m$^2$) | −2.0359 ± 1.8813 | −3.6000 ± 1.8670 | −1.1713 ± 1.4105 |

*3.2. Machine Learning Model Results*

The CNN part of our CNN-RNN model contains two blocks (the value of N shown in Figure 3 is 2). Each 1D convolutional operation contains 100 filters, with a kernel size of 3 and a padding of 1. While the number of LSTM units in the RNN is 200. We train our CNN-RNN model by exploiting Adam's method in [21] to minimize the mean square error (MSE) loss based on the training dataset. The evolution of the loss values to the training epoch of the CNN-RNN model is shown in Figure 5 (left), from which we can see that the loss value quickly converges to the minimum after several training epochs by Adam's method. The trained CNN-RNN model can then be used for yield prediction. As shown in Figure 5 (right), the predicted yield values by the trained CNN-RNN model are close to the ground truth ones.

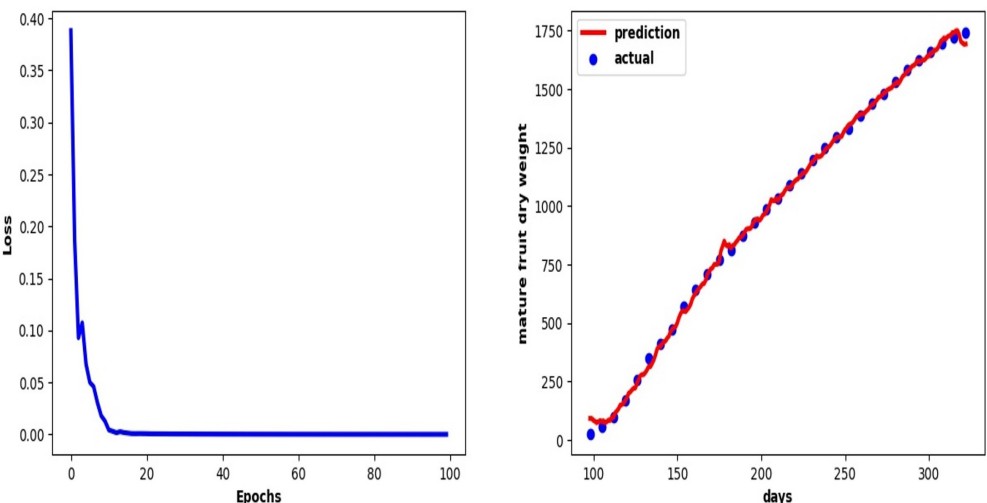

**Figure 5.** The evolution of the loss function with respect to the training epoch (**left**). The prediction results based on the trained CNN-RNN (**right**).

The CNN-RNN model is trained based on the training dataset, and we calculate different metrics between the ground truth tomato yields and the CNN-RNN model predicted ones based on the testing dataset. While the calculated mean and std of multiple RMSEs, r2 coefficients, NSEs and Pbiase values are shown in Table 4. Moreover, we also provide the performance of other classical machine learning models (multiple layer

perceptron (MLP), support vector regressor (SVR), random forest regressor (RFR) and gradient boosting regressor (GBR) as in [15]) and deep learning counterparts (LSTM-RNN in [23] and CNN), from which we can find that the CNN-RNN-based approach can achieve the best performance with the smallest mean RMSE and PBIAS absolute values and the largest R2 and NSE values.

**Table 4.** Statistical metrics (mean and standard deviation) of different metrics for tomato fruit yield prediction by different methods on the testing dataset.

|  | **RMSEs** | **R2s** | **NSEs** | **PBIASE** |
| --- | --- | --- | --- | --- |
| MLP | $66.63 \pm 17.01$ | $0.9863 \pm 0.0064$ | $0.9863 \pm 0.0066$ | $0.7465 \pm 1.9304$ |
| SVR | $116.78 \pm 0$ | $0.9318 \pm 0$ | $0.9330 \pm 0$ | $6.1768 \pm 0$ |
| RFR | $26.73 \pm 0.33$ | $0.9979 \pm 0.0001$ | $0.9979 \pm 0.0001$ | $0.5817 \pm 0.0380$ |
| GBR | $26.70 \pm 0.57$ | $0.9980 \pm 0.0001$ | $0.9980 \pm 0.0001$ | $0.5227 \pm 0.0589$ |
| LSTM-RNN | $29.95 \pm 2.27$ | $0.9973 \pm 0.0005$ | $0.9973 \pm 0.0005$ | $0.0824 \pm 0.2260$ |
| CNN | $31.26 \pm 4.99$ | $0.9961 \pm 0.0019$ | $0.9961 \pm 0.0019$ | $-0.1345 \pm 0.1810$ |
| CNN-RNN | $21.69 \pm 3.10$ | $0.9986 \pm 0.0004$ | $0.9987 \pm 0.0004$ | $-0.0363 \pm 0.5918$ |

### 3.3. Model Fusion Results

We have evaluated different combination approaches to fuse the outputs of the reduced Tomgro model and machine learning model to generate the final prediction output, which includes linear combination, Bayesian combination, neural network (NN)-based combination, random forest regressor (RFR)-based combination and gradient boosting regressor (GBR)-based combination. The performance of different approaches is presented in Table 5. From the results, we can see that: (i). most of the model fusion-based approach achieves more accurate performance than using the biophysical model or machine learning model solely with smaller mean RMSE and larger R2/NSE values (ii). Among different model fusion approaches, the neural network-based one achieves the best performance with the best RMSE, R2 and NSE metrics being obtained.

**Table 5.** Statistical metrics (mean and standard deviation) of different metrics for tomato fruit yield prediction by different methods on the testing dataset.

|  | **RMSEs** | **R2s** | **NSEs** | **PBIASE** |
| --- | --- | --- | --- | --- |
| Biophysical model | $36.17 \pm 7.64$ | $0.9980 \pm 0.0001$ | $0.9961 \pm 0.0016$ | $-1.1713 \pm 1.4105$ |
| CNN−RNN | $21.69 \pm 3.10$ | $0.9986 \pm 0.0004$ | $0.9987 \pm 0.0004$ | $-0.0363 \pm 0.5918$ |
| Linear combination | $20.85 \pm 3.19$ | $0.9992 \pm 0.0002$ | $0.9985 \pm 0.0005$ | $-0.1669 \pm 0.4103$ |
| Bayesian combination | $21.51 \pm 3.79$ | $0.9991 \pm 0.0003$ | $0.9982 \pm 0.0006$ | $0.4259 \pm 0.8575$ |
| NN combination | $17.69 \pm 3.47$ | $0.9995 \pm 0.0002$ | $0.9989 \pm 0.0004$ | $0.1791 \pm 0.6837$ |
| RFR combination | $18.68 \pm 2.94$ | $0.9994 \pm 0.0002$ | $0.9988 \pm 0.0003$ | $-0.1465 \pm 0.6758$ |
| GBR combination | $20.67 \pm 3.10$ | $0.9992 \pm 0.0002$ | $0.9985 \pm 0.0005$ | $-0.0715 \pm 0.7245$ |

## 4. Conclusions

In this work, we have developed a model fusion approach for fusing the outputs of both the biophysical model and machine learning model for predicting the tomato yield inside a greenhouse. The experimental results have shown that the model combination approach achieves better yield prediction results than solely using a biophysical or machine learning model. While the neural network-based model fusion approach achieves the best yield prediction results among all the methodologies, with the mean and standard deviations of RMSEs, R2s and Nash–Sutcliffe efficiency (NSE) being $17.69 \pm 3.47$,

0.9995 ± 0.0002 and 0.9989 ± 0.0004 respectively. The developed model can be applied to the greenhouse condition for the greenhouse crop yield prediction purpose, which can predict the future crop yield based on both greenhouse climate information and historical crop yield information.

In future work, we will evaluate the effectiveness of the developed technique for yield prediction for different crops (e.g., potato, strawberry, etc.) not only limited to tomato at different growing sites. Moreover, more advanced machine learning (e.g., transformer) or biophysical models will be investigated to be incorporated into the proposed model fusion framework targeting, achieving more accurate crop yield prediction. We will also try to incorporate the more accurate yield prediction algorithm into the greenhouse control system to achieve more precise greenhouse control to guarantee the maximum crop yield.

**Author Contributions:** Writing—original draft preparation, L.G.; methodology development and validation, L.G.; writing—review and editing, M.Y., V.C., S.K. and S.P. All authors have read and agreed to the published version of the manuscript.

**Funding:** This research was supported as part of SMARTGREEN, an Interreg project supported by the North Sea Programme of the European Regional Development Fund of the European Union.

**Data Availability Statement:** Data are available on request.

**Conflicts of Interest:** The authors declare no conflict of interest.

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
