# Peer review of "A Novel Model Fusion Approach for Greenhouse Crop Yield Prediction"

_horticulturae, doi:10.3390/horticulturae9010005_

Round 1

Reviewer 1 Report

This research investigated the possibility of using dee  neural networks for predicting tomato yield in greenhouse conditions. The research subject is interesting to the relevant readers; however, I found many formatting issues and spelling mistakes need to be corrected, for example: 

Line 26.....ny facilitating .... what is "ny"

Line 40: in  [7].... the authors need to use surname instead of number. Similar problems are found throughout the manuscript.  

The authors need to address the above issues before considering the acceptance of the manuscript for publication.  

Reviewer 2 Report

Authors developed a novel fusion-based approach, combining outputs from a biophysical and a machine learning models for a more accurate prediction of tomato yield in a greenhouse. Experimental results have shown that the fusion model approach achieves more accurate prediction results than the explanatory biophysical model or the machine learning model. However paper has limited novelty.

1.       The novelty contributions are limited.

2.       Reduce the abstract and also add some quantitative findings in the abstract.

3.       Authors should explain the equation 1 by providing suitable reference.

4.       Reduce the introduction section by reducing the irrelevant contents.

5.       Figure 2 is poorly drawn and presented.

6.       More comprehensive comparisons are required.

7.       Authors should clearly define the existing equations by adding suitable citation to them. Difficult to understand novel or modified equations proposed by the authors.

8.       Conclusion should be reduced. Also, add some quantitative findings in the conclusion along with the suitable future work.

Reviewer 3 Report

The paper presents the combination of biophysical and machine learning models to enhance crop yield prediction of tomato plants grown in the greenhouse. The author employed the results from the optimised biophysical Tomgro model and CNN+RNN deep learning model to train a machine learning model for tomato yield prediction. They report that using a neural network provided the best prediction accuracy. 

Generally, the paper has valuable contributions and is well-written. It would however benefit expanding the results and conclusion segments.

I would like to suggest the authors to expand on their conclusion section to include future directions of this research. 

Also, the authors do not currently offer a direct comparison of their results to previous studies. How does the RMSE obtained compare to what has been previously published?

Below some minor suggestions are offered:

Multiple lines: Throughout the manuscript, the authors use “as in” to introduce the citation. Since the citation is in number format, it makes the phrase look incomplete and I recommend the authors to remove the “as in” expression and simply add the citation in the end.

Please review the manuscript for double spaces and small typos such as in lines 79 and 130.

Line 25: It has a word segment that seems to belong to a deleted phrase: “ny”.

Reviewer 4 Report

The study is a methodolgy paper to compare the accuracies of mulitple combinations of models to predict the totamo yield.

Author should describe detail on models that were used in this study. please write modeling background sections and make separate tables to present the key components that were used in the model. It would be nice, if author provide graphic modeling framework. 

Author should calculate NSE, R2, PBIAS not only RMSE

in Absract, please write some result values. 

 Author found the best combinations of models to predict the tomato yields. But, author should use the found approach to simulate some meaningful results in some created scenarios. not just talk about the best approach.

Round 2

Reviewer 4 Report

Author should write about application of developed model to what condition in conclusion section. Author should metion about how the developed model can use in future study. 

Author Response

Thanks a lot for your comments. In the modified conclusion section, we have included both the application condition of developed model and how the developed model can use in future study.

Round 3

Reviewer 4 Report

none